# Panel Sequencing for Clinically Oriented Variant Screening and Copy Number Detection in Chronic Lymphocytic Leukemia Patients

**DOI:** 10.3390/diagnostics12040953

**Published:** 2022-04-11

**Authors:** Mariam Ibáñez, Esperanza Such, Alessandro Liquori, Gayane Avestisyan, Rafael Andreu, Ana Vicente, María José Macián, Mari Carmen Melendez, Mireya Morote-Faubel, Pedro Asensi, María Pilar Lloret, Isidro Jarque, Isabel Picón, Alejandro Pacios, Eva Donato, Carmen Mas-Ochoa, Carmen Alonso, Carolina Cañigral, Amparo Sempere, Samuel Romero, Marta Santiago, Guillermo F. Sanz, Javier de la Rubia, Leonor Senent, Irene Luna

**Affiliations:** 1Department of Hematology, Hospital Universitario y Politécnico La Fe, 46026 Valencia, Spain; such_esp@gva.es (E.S.); andreu_raflap@gva.es (R.A.); avicente_ana@gva.es (A.V.); macian_martor@gva.es (M.J.M.); melendez_mca@gva.es (M.C.M.); asensi_ped@gva.es (P.A.); lloret_marmad@gva.es (M.P.L.); jarque_isi@gva.es (I.J.); sempere_amp@gva.es (A.S.); romero_sam@gva.es (S.R.); santiago_marbal@gva.es (M.S.); sanz_gui@gva.es (G.F.S.); delarubia_jav@gva.es (J.d.l.R.); senent_leo@gva.es (L.S.); luna_ire@gva.es (I.L.); 2Hematology Research Group, Department of Medicine, La Fe Health Research Institute, University of Valencia, 46026 Valencia, Spain; alessandro_liquori@iislafe.es (A.L.); gayane_avetisyan@iislafe.es (G.A.); mireya_morote@iislafe.es (M.M.-F.); 3Centro de Investigación Biomédica en Red de Cáncer, CIBERONC, Instituto de Salud Carlos III, 28029 Madrid, Spain; 4Departamento de Ciencias Biomédicas, Facultad de Ciencias de la Salud, Universidad CEU Cardenal Herrera, 46115 Valencia, Spain; 5Department of Hematology, Hospital de Manises, 46940 Valencia, Spain; ipicon@hospitalmanises.es (I.P.); apacios@hospitalmanises.es (A.P.); 6Department of Hematology, Hospital Universitario General de Castelló, 12004 Castelló de la Plana, Spain; eva_donato@yahoo.es (E.D.); c.c.ortiz@hotmail.es (C.C.); 7Department of Hematology, Hospital Arnau de Vilanova, 46015 Valencia, Spain; mas_caroch@gva.es (C.M.-O.); alonso_carpri@gva.es (C.A.)

**Keywords:** CLL, NGS target panel, somatic variants and CNVs, prognostic and/or predictive genes

## Abstract

According to current guidelines, in chronic lymphocytic leukemia (CLL), only the *TP53* molecular status must be evaluated prior to every treatment’s initiation. However, additional heterogeneous genetic events are known to confer a proliferative advantage to the tumor clone and are associated with progression and treatment failure in CLL patients. Here, we describe the implementation of a comprehensive targeted sequencing solution that is suitable for routine clinical practice and allows for the detection of the most common somatic single-nucleotide and copy number variants in genes relevant to CLL. We demonstrate that this cost-effective strategy achieves variant detection with high accuracy, specificity, and sensitivity. Furthermore, we identify somatic variants and copy number variations in genes with prognostic and/or predictive value, according to the most recent literature, and the tool provides evidence about subclonal events. This next-generation sequencing (NGS) capture-based target assay is an improvement on current approaches in defining molecular prognostic and/or predictive variables in CLL patients.

## 1. Introduction

Chronic lymphocytic leukemia (CLL) is a markedly heterogeneous disease caused by the clonal expansion of mature B lymphocytes, causing impaired apoptosis and increased cell proliferation. Recently, the development of novel oral targeted agents has achieved better survival rates of CLL, improving the quality of life of the patients. This new paradigm affects the value of prognostic and/or predictive biomarkers that guide therapeutic options [1]. The mutation status of the immunoglobulin heavy-chain variable region (IGHV) genes and the deletions of 13q, 11q/*ATM/BIRC3*, and 17p/*TP53* are the factors with the utmost clinical impact regarding survival and outcome after treatment [2].

However, in recent years, consecutive multi-center studies have outlined a wide profile of genetic alterations, many of them recurrent, affecting several cellular pathways with potential clinical relevance [3,4]. The most frequently affected genes included *TP53*, *SF3B1*, *NOTCH1*, *ATM*, and *BIRC3* (unfavorable), *MYD88* (favorable), and *BTK*, *PLCG2* and *BCL2* (targeted drug resistance). Together with IGHV status, these variants allow for the establishment of a prognostic algorithm according to a significant interaction pattern of incidence, co-occurrence, and/or mutual exclusivity [5,6]. For many of these interactions, specific therapeutic approaches have been developed, with efficiencies directly related to the particular molecular characteristics of the patient [5,6,7,8,9,10]. Likewise, it seems that the consecutive acquisition or selection of tumor subclones can trigger the progression to a more aggressive CLL, with treatment resistance or final transformation. Therefore, next-generation sequencing (NGS) approaches are essential to detect simultaneously these genetic variants. In addition, NGS assays allow for the detection of minor mutated clones, especially relevant in *TP53*, *BTK*, *PLCG2*, and *BCL2*, although their clinical significance still remains controversial and requires further investigation [6,11,12,13].

Despite the knowledge of this genomic landscape, according to current international CLL guidelines, only IGHV and *TP53* molecular status are mandatory prior to every treatment decision, being especially relevant the report of *TP53* status [10,14,15,16]. However, additional heterogeneous genetic events confer a proliferative advantage to the tumor clone and are associated with progression and treatment failure in CLL patients. Thus, mutational profile assessment might translate into improvements in the clinical setting.

Therefore, in this study, we aimed to implement into daily clinical practice a single-run NGS strategy for the simultaneous detection of somatic variants and related copy number variations (CNVs) in the most relevant genes regarding a prognostic and/or predictive role in CLL. In addition, we assessed the ratio of patients with subclonal driver mutations within these genes that might trigger the clonal evolution of CLL.

## 2. Materials and Methods

### 2.1. Patient Material

A total of 119 consecutive untreated CLL patients were included and followed-up between October 2019 and 2021. Diagnoses were established according to the International Workshop on Chronic Lymphocytic Leukemia (iwCLL) guidelines and 2016 WHO revision [14,16,17,18]. Written informed consent was obtained in accordance with the Declaration of Helsinki, as well as approval from the review board of Bioethics and Medical Research at IIS La Fe.

Genomic DNA from patients was isolated from peripheral blood (PB) obtained before initiating therapy using a QIAamp^®^ DNA Mini Kit (QIAGEN, Hilden, Germany). Nucleic acids’ quantity/quality was assessed by the Qubit dsDNA High Sensitivity Kit and Nanodrop (Thermo Scientific, Waltham, MA, USA).

### 2.2. Conventional Cytogenetics

For copy number analysis, interphase fluorescence in situ hybridization (FISH) studies were performed on PB according to standard procedures; results were reported according to the International System for Human Cytogenetic Nomenclature (ISCN, 2020 recommendations). FISH probes included regions such as 11q22/ATM, 12q13, 13q14, and 17p13/TP53 (Vysis/Abbott Co., Downers Grove, IL, USA).

### 2.3. Library Preparation and Sequencing

In collaboration with SOPHiA genetics^®^, we designed a single NGS capture-based target enrichment assay for the simultaneous detection of single-nucleotide variations (SNVs), including small insertions and deletions (indels), and CNVs in 15 genes recommended by the European Research Initiative on CLL (ERIC) [(Complete coding and UTR regions: *ATM*, *BIRC3*, *CXCR4*, *EGR2*, *FBXW7*, *KRAS*, *MYD88*, *NFKBIE*, *POT1*, and *TP53*; Hotspot regions: *BTK*, *NOTCH1*, *PLG2*, *SF3B1*, and *XPO1*)] and CNVs [(Detection limit of 20%): ATM deletion, trisomy of chromosome 12 [*KRAS* (12p12), *ATF1* (12q13), and *CDK4* (12q14)], deletion 13q/monosomy 13 [*DLEU* (13q14), *PROZ* (13q34), *KLF5* (13q22), and *CUL4A* (13q34)] and *TP53* (17p12)].Targeted enriched library preparation was carried out following the manufacturer’s procedures. Sequencing was performed on a V3 flow cell using a MiSeq sequencer (Illumina), according to the manufacturer’s procedures.

### 2.4. Bioinformatics Data Analysis

Fastq sequence files were uploaded in the SOPHiA DDM^®^ analytical platform and the SOPHiA GENETICS proprietary workflow was applied. Reads were pre-processed and aligned to the human genome of reference (hg19). Then, the pipeline performed the identification of the different types of variants (either single-nucleotide variant, indel, or copy number), and finally the annotation based on clinical databases, differentiating between retained and low-confidence variants.

SNV and indel variants found in coding regions and ± 10 bp to splice site regions were considered when their minor allele frequency (MAF) in ClinVar, 1000 Genomes Project, ExAC, and/or dbSNP was smaller than 0.01. The predicted functional consequences on splicing and on the protein (excluding synonymous variants unlikely damaging) of variants at variant allele frequency (VAF) clonal ≥10% (i.e., clonal) and <10% (i.e., subclonal) were also assessed. Finally, variants were reported using the international standard HGVS nomenclature and categorized into 5 categories: pathogenic (P), likely pathogenic (LP), variant of uncertain significance (VUS), likely benign (LB), and benign (B), according to the American College of Medical Genetics and Genomics (ACMG) criteria, IARCTP53, *TP53* WEBsite and Palomo et al., 2020 [17]. Putative pathogenic (P), likely pathogenic (LP), and VUS variants were manually inspected in the Integrative Genomics Viewer (IGV).

For CNVs, the manufacturer developed a secondary module, based on intra-run sample normalization, in terms of region coverage. For this reason, a minimum of 8 samples, simultaneously performed in the wet lab part, were analyzed together, as a batch. By default, the module offered CNV estimation for each region included in the panel, rejecting those where the uniformity did not allow a final conclusion to be provided. Finally, for the particularity of this panel (with high-frequency CNV regions), SOPHiA GENETICS specifically created an extra script, adapting the comparison parameters, in order to infer alterations on these specific regions.

### 2.5. NGS Design Validation

As a set-up program, inter-run reproducibility and intra-run repeatability were determined by the analysis of 18 unique samples and 2 reference samples (SG063 and HD701 references) previously characterized by the NGS-gen approach CLL MSTR Plus Kit^®^ and by FISH. Two independent runs of 32 different libraries each (4 captures of 8 samples), including intra-/inter capture and intra/inter-run replicates, were performed. Samples of this study with known mutations in *TP53*, *SF3B1*, *NOTCH1*, *CXCR4*, and *MYD88* genes were confirmed, previously validated by other high-confidence techniques, such as Sanger sequencing and/or ASO-PCR, as previously reported [18,19].

Accuracy, sensitivity, and specificity parameters were assessed as follows: accuracy = (true positive (TP) + true negative (TN))/(TP + false positive (FP) + TN + false negative (FN)); sensitivity = TP/(TP + FN); specificity = TN/(TN + FP).

## 3. Results

### 3.1. Analytical Data of Genomic Variants

The library sequencing yielded an average of 43 million readings per run, 98% of which successfully mapped to the human reference genome. Among these, 81% were mapped on the regions and/or flanking regions of interest included in the design, while 19% were off-target, mainly found in pseudogenes or scattered genome-wide. In addition, we observed 98.95% of target regions with high on-target coverage (>1000 readings per position), the recommended minimum coverage for having confident results with SOPHiA’s algorithm. Finally, on average, coverage uniformity resulted in 99.22% of the panel, which supports the consistency of the assays. 

The concordance rate for the detection of SNVs and indels was 91%. Among 20 samples, 103 SNVs or indel variants present at VAF > 3% were confirmed (true positive). Additionally, a high concordance of VAF was observed between both assays. In total, no false negative results were detected. However, 17 variants were detected by SOPHiA DDM^®^ in characterized regions not previously reported (false positive, FP). These FP variants had low variant fractions and occurred in low-complexity or homopolymer regions. Finally, 132 additional variants were reported, previously described in public databases (dbSNP, COSMIC). Based on all the detected variants, the concordance between replicates of the intra and inter-run replicates was >98.9%, accuracy and specificity were 86%, and sensitivity was 100%.

### 3.2. Somatic Variants Landscape

The CLL sequencing panel detected a total of 2804 full variants among the 119 patients analyzed. After applying filters, 363 variants detected in 14 out of 15 genes, 185 as unique variants, were further considered relevant. Mean read depth was 3578× (range: 483×–9797×), and mean VAF was 18% (range: 0.1–89). Among them, 80% were SNVs (*n* = 289) and 20% were indels (*n* = 74). According to their functional classification, 234 variants were missense, 65 frame-shift, 3 nonsense, 5 start-loss, 1 stop-loss, 1 synonymous, 34 were located in the 3′-UTR region, and 10 in splice acceptor or donor sites. At least one point variant was identified in 95 patients (83%), with a mean of 3.8 mutations by the patient (range: 1–17). Once categorized, 153 variants were classified as pathogenic (P), 79 likely pathogenic (LP), and 44 of uncertain significance (VUS), being 90, 58, and 37 unique variants, respectively (Figure 1A). As expected, *TP53* exhibited the greatest number of variants (*n* = 60), followed by *SF3B1* (*n* = 31), *NOTCH1* (*n* = 30), *ATM* (*n* = 29), and *BIRC3* (*n* = 29) (Figure 1B and Figure 2).

Variants detected in the clonal-cell population (VAF ≥ 10%) represented 41% of the total count (*n* = 159), and 54% of them were unique variants (*n* = 86). Seventy variants had been previously reported in CLL and categorized as pathogenic, while 32 were classified as likely pathogenic and 19 as VUS, of which 41, 26, and 18, respectively, were unique. Clonal variants’ incidence ranged from 1 to 8 (average 1.8), with 59% of the patients (*n* = 67) having at least one mutation identified in 14 of the 15 screened genes. These mutations commonly occur in *TP53* (21%), *NOTCH1* (17%), *ATM* (13%), and *SF3B1* (12%). Interestingly, in 12 of the patients (18%), we found genes harboring multiple mutations in the same gene (range 2–3), affecting 4 of the 14 genes, which were *TP53*, *ATM*, *SF3B1*, and *POT1.* Furthermore, at least one additional subclonal variant was detected in 40 patients (average: 2–7; range: 1–8), involving, mainly, *NFKBIE* and *TP53* genes.

Subclonal variants (VAF < 10%) in potentially actionable genes, such as *TP53*, *SF3B1*, *NOTCH1*, and *BTK*, were present in 64 patients (56%), 22 of them without additional clonal variants. In addition, 38 patients carried more than one subclonal variant at the same time. Overall, 83 variants were considered pathogenic, 46 likely pathogenic, and 25 VUS, being the most frequent those found in genes *TP53*, *NFKBIE*, and *BIRC3*. Finally, there were no differences in the distribution of the variant categories based on the VAF (≥10%; <10%) (Figure 1A). However, remarkably, subclonal variants displaying VAF = 5–9.9% mainly affected *TP53*, *SF3B1*, *NOTCH1*, or *BTK* genes and were observed in 24 patients (21%) with a mean coverage of 3913× (range: 2726–6965).

### 3.3. TP53 Variants 

In 39 patients (41%), we detected 63 variants categorized as pathogenic, likely pathogenic, or VUS in *TP53*. Among the 54 unique variants identified in exons 4 to 10, 42 variants were missense (6 clustered in the common hotspots [R273 (*n* = 3), R248 (*n* = 2), and Y220 (*n* = 1)]), 6 affected the splice-site regions, 4 were frameshift and 2 nonsense (Figure 2). On average, the coverage was 3300× (range: 950–6571) with a median VAF of 19% (range 1–85). Over half of the variants displayed VAF > 10% in 24 patients. However, in 21 patients, at least one subclonal variant was detected with an average of 3% of VAF and with more than 3600× of mean coverage. These variants were mainly missense and were mostly located in exons 5, 7, and 8.

Considering patients with available cytogenetics paired data (*n* = 70), patients with alterations in *TP53* were regrouped in five sets as follows: (a) a single *TP53* mutation without chromosomal abnormalities affecting the *TP53* locus (*n* = 15, 21%); (b) multiple mutations without del17p12 (*n* = 11, 16%); (c) any number of mutations with concomitant chromosomal abnormalities affecting the *TP53* locus (*n* = 9, 13%); (d) del17p12 and *TP53^wt^* (*n* = 3, 4%), and (e) no chromosomal abnormalities involving the *TP53* locus and *TP53^wt^* (*n* = 32, 46%). Finally, according to the number of *TP53* genomic lesions, patients were categorized into three different *TP53* allelic states: mono-allelic (group a, (*n* = 15, 21%)), multi-hit (group b and c; (*n* = 20, 29%)), and *TP53^wt^* (group d and e, (*n* = 41, 50%). However, the follow-up period was not long enough to provide statistical significance.

### 3.4. Detection of Copy Number Variations 

Subsequently, we performed copy number estimation using the SOPHiA DDM^®^ analytical platform and compared the obtained results with fluorescence in situ hybridization (FISH) results when possible. NGS confirmed 78% of the CNVs identified by conventional cytogenetic, detecting a high correlation for positive predictive values (84%). However, most likely biased due to a high density of cases carrying the same CNVs in the same run, the negative predictive value for del13q detection was significantly lower (50%).

We observed CNVs in 52 patients, being del13q (*n* = 33) the most frequent. Among these patients, 61% displayed del13q as a single alteration, lacking variants in the screened gene panel. The same was true for three additional patients with CNVs affecting chromosomes 11 or 12. The most frequent co-occurring variants involved *TP53* and *SF3B1* genes. Finally, 81% of the patients with del17p harbored clonal and/or subclonal variants simultaneously.

## 4. Discussion

This study shows a single-center experience developing and validating a comprehensive and up-to-date single NGS-based screening for the simultaneous detection of the most clinically relevant SNVs and CNVs on a consecutive cohort of patients with CLL. Among the 119 untreated CLL patients, a total of 363 variants were detected in 83% of the patients, identified in 14 of the 15 screened genes. After an accurate categorization, 76% of the detected variants were classified as pathogenic, likely pathogenic, or VUS and further considered. On average, patients carried almost two mutations per sample, affecting mainly well-known genes related to CLL: *TP53*, *SF3B1*, *NOTCH1*, *ATM*, and *BIRC3*. Frequencies of the variants agreed with those previously reported [1]. In addition, subclonal variants in potentially actionable genes were detected in 64 patients, 34% of them lacking concomitant clonal variants. Variant categorization followed the same distribution as the clonal variants, involving *TP53*, *NFKBIE*, and *BIRC3*. However, subclonal variants with VAF values 5–9.9% were observed in 24 patients and affected *TP53*, *SF3B1*, *NOTCH1*, or *BTK* genes.

An improved understanding of the pathogenesis of CLL has provided a large number of genetic alterations, many of them recurrent, with potential clinical relevance, but also clues regarding the outline of clonal evolution triggering an initially indolent and stable disease into more aggressive states [3,4]. These molecular alterations show significant patterns of incidence, co-occurrence, and/or mutual exclusivity, hence allowing for the establishment of a predictive and prognostic algorithm [5,6,9,10]. Current chemoimmunotherapy strategies and biological drugs rely upon specific molecular features, with efficiencies directly related to the genome profile of the patient [7,8]. As a result, the number of clinically relevant gene mutations identified in CLL has increased dramatically. Although current international guidelines only recommend assessment of the mutational status of *TP53* and IGHV prior to each line of treatment [5,6,7,8,9,10,14,16], during the last few years, high-throughput technology approaches have been developed and used in order to screen, simultaneously, a broad spectrum of CLL-related genes. Recently, standard practice recommendations were published by ERIC for the assistance of laboratories when introducing targeted NGS into the laboratory [11,12]. Critical parameters such as coverage, sensitivity, and reproducibility have been established. In concordance with these guidelines, our capture-based approach reached good quality, with a high number of read counts per sample, a successful capture with very few low-coverage regions, and excellent overall intra-run and inter-run reproducibility, sensitivity, and specificity.

NGS approaches permit the more reliable detection of low-level variants. In our study, more than half of the patients presented subclonal variants, affecting, mainly, *TP53*, *NFKBIE*, and *BIRC3* genes. Although ERIC guidelines recommend reporting variants with a minimum of 10% VAF, a clinically significant VAF cut-off is not yet well determined [16]. Low-level variants in genes with diagnostic, prognostic, and/or predictive implications may be clinically relevant. However, this is not only true for *TP53*, where clonal expansion has been widely described at relapse, but also for specific mutations in *BTK* or *PLCG2* and *BCL2*, associated with disease progression and subclonally detected many months before clinical progression [6,19]. Although, currently, the subclonal mutation information does not influence the choice of therapy, it is important to identify which patients would require closer observation. However, standardizing technical and interpretation aspects to achieve greater sensitivity and guarantee the consistent and accurate detection of low-frequency variants is challenging [20]. In this sense, a recent ERIC initiative demonstrated that minor subclonal variants (VAF < 5%) displayed lower reproducibility and worse precision. Thus, a VAF cut-off of 5% or 10%, widely used in clinical routine, could be considered [11,12]. In our study, variants bordering the 5–10% threshold were present in 24 patients and involved the *TP53*, *SF3B1*, *NOTCH1*, and *BTK* genes. Recently, ERIC led a multi-center project specifically focusing on low-frequency *TP53* mutations to establish a cut-off for reporting variants [13]. Meanwhile, in patients with subclonal variants, tumor infiltration should be considered and evolutionary studies should be performed.

Of note, our NGS-based approach also allows for the detection of clinically actionable CNVs, and the obtained results were further compared with FISH results. In general, we confirm good levels of sensitivity in most of the analyses, with only a lower concordance for del13q detection. Discrepancies might be a natural consequence of analyzing highly prevalent changes with a tool-based intra-run comparison of sample coverage. However, this module, secondary to SNV/InDel detection, was not claimed to totally replace the FISH analysis, but to complement it. In addition, non-detected alterations were below the sensitivity threshold (20% of cells). These findings are in agreement with those previously reported for the reliable detection of deletion or duplication by NGS [18,20,21].

This novel NGS-based assay allows the screening of all clinically relevant genetic alterations for the diagnosis and prognostic risk stratification of patients with CLL. The identification of clinically important alterations at a sustainable time and cost promotes the integration of our NGS approach into the current diagnostic strategies in the short/medium term. Through this NGS system, many samples per run can be analyzed simultaneously with deep and homogeneous coverage and from a small amount of sample. Obtained results can be analyzed using a user-friendly analytical system, which allows the visualization of every variant, marking the retained ones, showing a pre-classification, as well as performing a complex filtering process with all databases linked, reducing significantly the interpretation time. Finally, an additional advantage of our NGS platform is that the assessment of the mutational state of several genes that ought to be considered in CLL can be performed in a single experiment. The first and most important gene to analyze by NGS is *TP53*, whose prognostic significance and response prediction have been seen in both clonal and subclonal mutations. In addition, Sanger sequencing has limited sensitivity to detect variants with a VAF below 20% (no longer only subclonal variants). Moreover, additional cost should be added in case the CD19+ population needs to be selected, in an attempt to improve the sensitivity limit. However, we cannot forget that the ERIC guidelines recommend the sequencing of almost the totality of the gene, from exons 2 to 10 or, at least, from 4 to 8. Sequencing eight exons by Sanger has already a cost comparable to the CLL gene panel [11,12,13]. In addition, the characterization of *BTK* and *PLCG2* is considered of importance during the progression of the disease due to their impact on therapeutic decision-making in CLL patients [6,11,12,13]. Lastly, regarding genes with proven prognosis involvement, such as *NOTCH1*, *BIRC3*, *MYD88*, or *SF3B1*, even though they are not yet established in the current practical guidelines, in the coming years, they might be included as useful tools for the follow-up management of CLL patients, perhaps through a closer follow-up for patients with poor prognostic mutations [5,6,7,8,9,10]. 

The main limitation of this study is the short-term follow-up data, although the goal of the study was not to analyze the association of variants and clinical–biological features, widely established in the literature. The additional drawback would be the need of confirming CNVs by FISH in highly frequent changes. Finally, variants present in minor subclones are frequently detected through this NGS approach. However, reliable validation of these variants as true positive requires complex, costly, and time-consuming additional approaches not always well implemented in a clinical laboratory. 

## 5. Conclusions

NGS approaches are essential to detect variants with VAF < 20%, very frequent in CLL patients and involving, mainly, the *TP53* gene. However, to achieve greater sensitivity and guarantee the consistent and precise detection of variants, the development of standardizing guidelines for technical aspects and variant interpretation is essential. This comprehensive approach covers somatic mutations and CNVs within significant driver genes in a timely and cost-effective manner and represents a significant improvement over current strategies in defining abnormalities in CLL. The clinical impact of subclonal variants on CLL is still controversial and should remain under research, requiring complementary studies in larger cohorts to establish their real role.

## Figures and Tables

**Figure 1 diagnostics-12-00953-f001:**
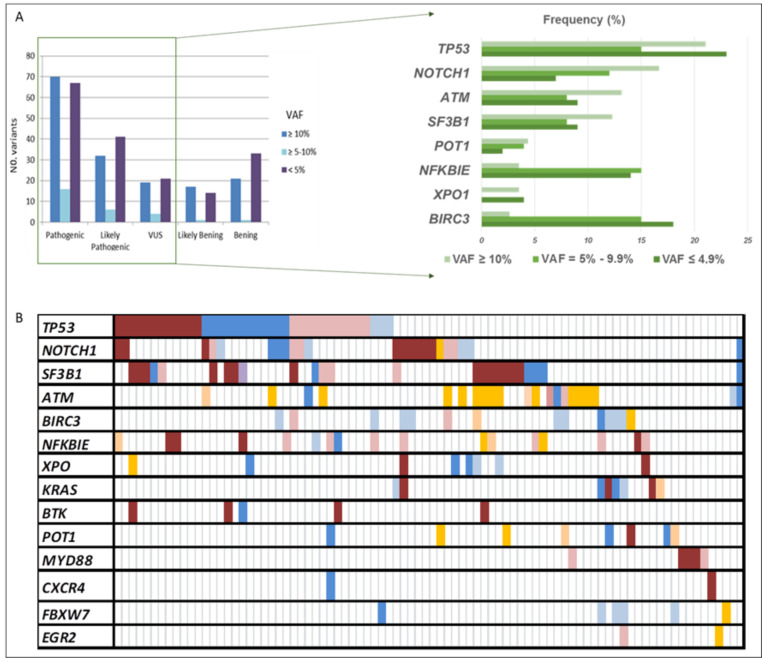
All variants’ distribution detected in our CLL cohort. (**A**) The number of variants according to their VAF and pathogenicity categorization. (**B**) Summary of the frequency distribution of variants stratified by gene name and variant pathogenicity categorization. Columns represent patients and rows represent genes. Color coding indicates the type of variant (red pathogenic, blue likely pathogenic, and orange uncertain significance), whereas the intensity depicts the variant allele frequency (darker corresponds to clonal variants; lighter to subclonal variants). VAF: variant allele frequency.

**Figure 2 diagnostics-12-00953-f002:**
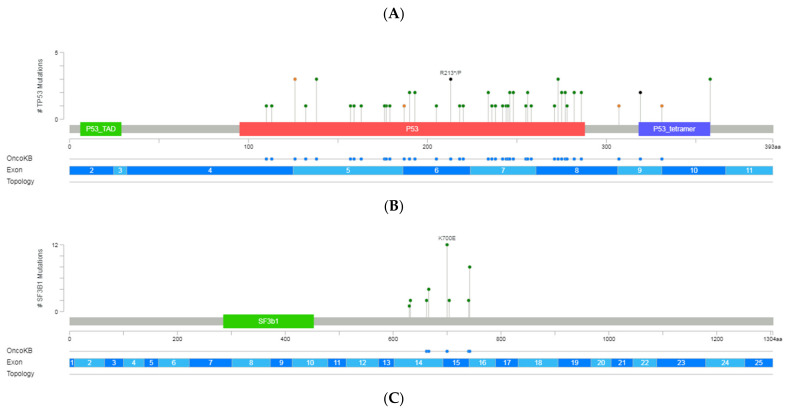
A schematic representation of variant localization across the most frequently mutated genes in our cohort: *TP53*, *SF3B1*, *NOTCH1*, *ATM*, and *BIRC3.* Circles, colored based on mutation types, represent variant position, and the length of the line is directly proportional to the number of variants detected at the same codon. On top, the most frequent variant is highlighted at the specific site. Green circle indicates missense mutations; brown circle, truncating mutations (nonsense, nonstop, frameshift deletion, frameshift insertion, and splice site); black circle, inframe mutations (inframe deletion and inframe insertion); purple circle, other mutations (all other types of mutations). When different variants are located at the same position, the color of the circle is determined by the most frequent variant. For each gene, different hotspot amino acid (aa) positions are displayed at grey bars and specific functional domains in colored boxes. (**A**) Mutations identified in the *TP53*, (**B**) *NOTCH1*, (**C**) *SF3B1*, (**D**) *ATM* and (**E**) *BIRC3* (Gao et al. Sci. Signal. 2013 & Cerami et al., Cancer Discov. 2012).

## Data Availability

Not applicable.

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
