# Peer review of "Panel Sequencing for Clinically Oriented Variant Screening and Copy Number Detection in Chronic Lymphocytic Leukemia Patients"

_diagnostics, 2022, doi:10.3390/diagnostics12040953_

Round 1
Reviewer 1 Report
In the manuscript „Panel Sequencing for Clinically Oriented Variant Screening and Copy Number Detection in Chronic Lymphocytic Leukemia Patients“ authors describe the implementation of a single NGS capture-based target enrichment assay to detect both SNVs and CNVS in patients with CLL. Analysis was performed on 119 patients and results are represented by all variant distributions in their CLL cohort and by a schematic of variant localization across the most frequently mutated genes. They disclose limitations of their study, the main one being short-term follow up.
Broad comments: The manuscript is logically written and reports on the important matter. However, I have some minor comments to improve the manuscript.
Specific comments:
- Please highlight in the discussion section the novelty of your study
- Figure 2 is difficult to read, please improve the resolution and enlarge the font (exon, topology, gene names…)
- Figure 1 - VAF should we written in full also
- Line 91 – explain the indels meaning
- Line 166, you have already explained VUS abbreviation in section 2.4
Author Response
Point 1. Please highlight in the discussion section the novelty of your study
Response 1: Please find a new paragrah at discussion section. See Discussion, paragraph 5, lines 1-5
Point 2. Figure 2 is difficult to read, please improve the resolution and enlarge the font (exon, topology, gene names…)
Response 2: Done.
Point 3. Figure 1 - VAF should we written in full also.
Response 3: Done.
Point 4. Line 91 – explain the indels meaning
Response 4: Done.
Point 5. Line 166, you have already explained VUS abbreviation in section 2.4
Response 5: Done.
Finally, our manuscript has been checked by a native English speaker and some changes have been applied.
Reviewer 2 Report
With the actual description, the article seems to miss one of the primary points. In the article, there are no evidence that this analysis could improve the prognosis of the patients. There is no data about how easy or feasible is to use this kind of analysis for medical center (costs and time). The article has no data about the follow-up of the patients, or an association regarding mutations and prognostic factors (for example overall survival or response to treatment). This is explained in the discussion, but this makes not clear if the mutations identified could be useful for prognostic use.
The article shows efficiently that this analysis able to detect variants with high accuracy but should at least add some data about that this analysis is a cost-effective strategy, or that add something useful as prognostic methodology.
Major corrections
Figure 2: the quality of the figure is too low to be read in electronic form. Moreover, the legend is not well written not explaining the meaning of the different colors and reporting some extra references in the wrong format (rows 183-184).
Minor corrections
Typos in rows: 52, 57, 60, 75, 179, 195, 219, 222 and 324.
Author Response
Response to Reviewer 2 Comments
Major corrections
Point 1. Figure 2: the quality of the figure is too low to be read in electronic form. Moreover, the legend is not well written not explaining the meaning of the different colors and reporting some extra references in the wrong format (rows 183-184).
Response 1. Done
Minor corrections
Point 1. Typos in rows: 52, 57, 60, 75, 179, 195, 219, 222 and 324.
Response 1. Done
In addition, taking into account your comment, a new paragraph has been included at discussion section about how our approach add something useful as diagnostic and prognostic methodology. Please, see Discussion, paragraph 5
Finally, our manuscript has been checked by a native English speaker and some changes have been applied.
Reviewer 3 Report
The study is comprehensive and the methods seem sound and reliable.
However, some issues should be addressed before publication:
line 118: batch analysis and comparison to batched mean values will likely lead to missing CNV that occur in a large portion of samples (eg del13q). You correctly addressed this issue in line 229 and in the Discussion.
line 158: it is not clear to me which filters were applied here
line 162: only 1 synonymous mutation out of 363 seems very low. were synonymous already filtered out before (line 158)?
line 196: what are potentially actionable genes?
line 202: please compare the read depth at these positions to exclude that they allow for a higher AF sensitivity when compared to other variants positions
line 211: as in the preceding comment, please check if differences in coverage affect increased variant calling sensitivity
Author Response
Response to Reviewer 3 Comments
Point 1. line 118: batch analysis and comparison to batched mean values will likely lead to missing CNV that occur in a large portion of samples (eg del13q). You correctly addressed this issue in line 229 and in the Discussion.
Response 1. Ok
Point 2. line 158: it is not clear to me which filters were applied here
Response 2. Further details have been added. Please see Material and Methods, Bioinformatics Data Analysis, 1st paragraph.
Point 3. line 162: only 1 synonymous mutation out of 363 seems very low. were synonymous already filtered out before (line 158)
Response 3: Done. Please note that only synonymous variants with protein impact have been retained.
Point 4. line 196: what are potentially actionable genes?
Response 4: Done. We have included those genes considered potentially actionable
Point 5. line 202: please compare the read depth at these positions to exclude that they allow for a higher VAF sensitivity when compared to other variants positions.
Response 5: Done. Median coverage data has been provided. In addition, the read depth at these positions has been checked and coverage uniformity resulted in > 99%.
Point 6. line 211: as in the preceding comment, please check if differences in coverage affect increased variant calling sensitivity
Response 6: Done.
Finally, our manuscript has been checked by a native English speaker and some changes have been applied.